# Low-Cost Inkjet-Printed Temperature Sensors on Paper Substrate for the Integration into Natural Fiber-Reinforced Lightweight Components

Johanna Zikulnig [1,*], Mohammed Khalifa [2], Lukas Rauter [1], Herfried Lammer [2] and Jürgen Kosel [1]

1 Silicon Austria Labs GmbH, Europastraße 12, 9524 Villach, Austria; lukas.rauter@silicon-austria.com (L.R.); juergen.kosel@silicon-austria.com (J.K.)
2 Wood Kplus—Kompetenzzentrum Holz GmbH, Klagenfurter Straße 87, 9300 St. Veit an der Glan, Austria; mohammed.khalifa89@gmail.com (M.K.); h.lammer@wood-kplus.at (H.L.)
* Correspondence: johanna.zikulnig@silicon-austria.com

**Abstract:** In a unique approach to develop a "green" solution for in-situ monitoring, low-cost inkjet-printed temperature sensors on paper substrate were fully integrated into natural fiber-reinforced lightweight components for which structural health monitoring is becoming increasingly important. The results showed that the sensors remained functional after the vacuum infusion process; furthermore, the integration of the sensors improved the mechanical integrity and stability of the lightweight parts, as demonstrated by tensile testing. To verify the qualification of the printed sensors for the target application, the samples were exposed to varying temperature and humidity conditions inside of a climate chamber. The sensors showed linear temperature dependence in the temperature range of interest ($-20$ to $60\,°C$) with a TCR ranging from $1.576 \times 10^{-3}\ K^{-1}$ to $1.713 \times 10^{-3}\ K^{-1}$. Furthermore, the results from the tests in humid environments indicated that the used paper-based sensors could be made almost insensitive to changes in ambient humidity by embedding them into fiber-reinforced lightweight materials. This study demonstrates the feasibility of fully integrating paper-based printed sensors into lightweight components, which paves the way towards integration of other highly relevant sensing devices, such as strain and humidity sensors, for structural health monitoring of smart, sustainable, and environmentally compatible lightweight composite materials.

**Keywords:** inkjet printing; temperature sensor; structural health monitoring; paper electronics; sustainable sensors

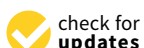



## 1. Introduction

Current social and ecological challenges demand innovative solutions for the efficient use of resources associated with the increased use of renewable raw materials. In the field of lightweight construction, natural fiber-reinforced biopolymers are a promising candidate for fulfilling the requirements for ecological compatibility while providing advanced material performance [1–3]. Natural fibers, such as flax, hemp, or cotton, have been extensively studied throughout recent years to evaluate their qualifications for mechanical reinforcement of lightweight parts [4–6]. In order to further improve the performance and reliability of the respective composites, structural health monitoring in the fields can be a powerful tool for future material development [7,8]. Printed sensors on biocompatible substrates, such as paper, are well-suited for the integration into the respective lightweight parts during manufacturing, as paper is a low-cost, easily available, sustainable, and biologically degradable material, and its application for printed electronics has been well-studied [9,10]. Speaking of sustainability, digital additive electronics manufacturing technologies, such as inkjet printing, are considered to have a low environmental impact provided that eco-friendly materials are used [11]. As illustrated in Figure 1, a sustainability cycle can be established: paper-based printed sensors and natural fiber-reinforced lightweight composites consist of sustainable materials and are manufactured in an ecologically friendly way.

By employing the sensors for structural health monitoring of the respective lightweight parts, valuable performance data from the fields will be collected, generating new knowledge, which can be used as a basis for improved material development to further enhance the mechanical properties and the longevity of sustainable lightweight composites.

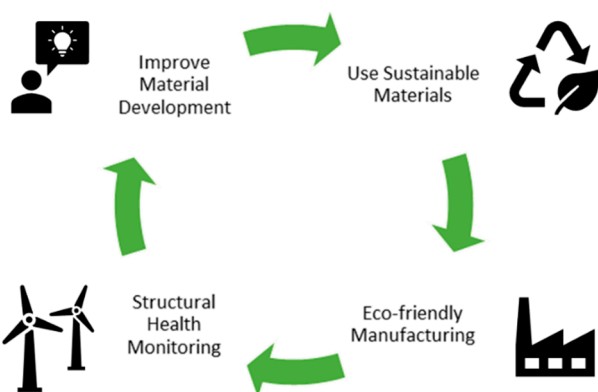

**Figure 1.** Cycle for sustainable material development enabled by low-cost resource-efficient sensors for structural health monitoring.

Printed sensors have been presented as promising candidates for structural health monitoring of large structures due to their low manufacturing costs, which consequently enable the cost-efficient integration of a large number of sensors over an extended area. As an example, Zymelka et al. [12] developed a screen-printed, temperature-compensated, graphite-based strain sensor array for the structural health monitoring of large areas. They impressively demonstrated the functionality of their devices by installing them on a highway bridge. Even 7 months and 1 year later, the sensors remained fully functional and were capable of detecting and localizing cracks in the monitored bridge accurately. In a different approach, Cook et al. [13] developed a passive inkjet-printed patch antenna sensor, which enables the detection of crack formation, orientation, and shape by means of resonant frequency shifts in the two resonant modes of the antenna. Zhang et al. [14] presented the development and manufacturing of inkjet and screen-printed strain sensors on polyethylene-terephthalate (PET) flexible substrates for the purpose of structural health monitoring in aircraft. While printed sensors have been proposed for structural health monitoring before [12–14], a full integration making the sensor an inherent part of the composite material to monitor, as presented in this work, is highly innovative and distinctive of previous works in this field.

The current case study builds upon the results presented in [15], where inkjet-printed resistive temperature sensors on commercial uncoated paper substrates have already been fabricated and characterized while evaluating their suitability for the proposed task of structural health monitoring of natural fiber-reinforced lightweight composites. The uncoated paper substrate does not contain any synthetic layers, which makes it fully ecological while being low-cost. However, considering the inkjet printing process, several challenges arise due to the highly porous, rough, and fibrous nature of uncoated paper compared to, e.g., most polymer-based substrates, resulting in lower conductivity, as well as reproducibility, and inhomogeneous layers [16,17]. On the other hand, the inkjet printer ink is partially absorbed, due to its low viscosity, increasing the adhesion and, consequently, the stability and durability of the printed films [18]. Results from [15] indicated that the bare sensors would be applicable for the task of structural health monitoring of natural fiber-reinforced lightweight materials, in particular rotor blades of small wind turbines. In the temperature range of interest for manufacturing of the lightweight parts (vacuum infusion process and thermal post-curing 20 to 80 °C), the sensors showed good linear temperature dependence, minimal hysteresis, and low baseline drift. In an extended temperature range (−25 to 150 °C) and when being exposed to humid environments (20 to 80% rH), the sensor performance worsened, which can mainly be attributed to fiber swelling, as the porous

paper substrate absorbs a large amount of ambient humidity, leading to a mechanical deterioration (cracking) of the printed structure [19].

Building on the results presented in [15], two paper-based temperature sensors were integrated into natural fiber-reinforced composites as part of the present work. After the integration, the samples were exposed to varying relevant temperature and humidity conditions inside of a climate chamber.

## 2. Materials and Methods

### 2.1. Inkjet-Printed Temperature Sensor

The sensors were fabricated in accordance with [15] using inkjet printing of Ag-nanoparticle ink (Sicrys 150-TM119, PVNanocell, Migdal Ha'Emek, Israel) on uncoated paper substrate. The sensors were designed as meander-line structures based on other resistive sensor designs reported in the literature [20–25] with a total sensing area of 45 × 25.5 mm as well as line width and spacing of 0.5 mm. A PIXDRO LP50 (Süss Microtec SE, Garching, Germany) system with a Spectra SE-128 AA 128 (Fujifilm Dimatix Inc., Santa Clara, CA, USA) 30 pL print head assembly at a resolution of 500 × 500 dpi was used. Two layers of ink had to be applied, as the first layer is largely absorbed by the paper fibers, which results in poor electrical conductivity, while the second layer on top of the first can be considered as comparatively homogeneous, as illustrated in Figure 2a,b for one and two printed layers, respectively. Between the two printing passes, no sintering was performed, as intermediate sintering can lead to interface structures in multilayer printing, which reduces the homogeneity of the conductive path [26]. After drying the two layers, the sensor structures were sintered. Due to the low thermal tolerance of the used paper substrate, photonic curing (Pulse Forge 1200, NovaCentrix, Austin, Texas. USA) at overall energy of 2.1 J/cm$^2$ was employed. As part of previous works, the sheet resistance of the same substrate-ink combination (2-layers) was evaluated using Van-der-Pauw's method [17,27]. According to this study, the median sheet resistance of the printed temperature sensors was expected to be around 60 mΩ/□, which equals to a specific resistivity ρ = 12 μΩ × cm (7.6 × bulk silver) at a layer thickness of 2 μm (as obtained by SEM imaging, see Figure 2c). However, this applied only in small regions, where the layer was rather homogeneous. As soon as a larger area was observed, the influence of the surface roughness and porosity of the paper substrate became obvious, which reduced the integrity of the printed Ag layer, as illustrated in Figure 2d.

Amongst other things, the electrical resistance of electrical conductors strongly depends on the temperature. For many materials, including silver, and in a specified temperature range, the relationship between the temperature T and the resistance R(T) can be approximated linearly using the following Equation (1) [28]:

$$R(T) = R_0 \times [\, 1 + \alpha \times (T - T_0)]  \tag{1}$$

$R_0$ equals the resistance at a defined temperature $T_0$ (as part of the present work, it was defined as $T_0$ = 20 °C). The material-dependent constant α corresponds to the temperature coefficient of resistivity (TCR), which is a positive value for metals (e.g., silver), implying that the sensor's resistance increases with increasing temperature.

### 2.2. Sensor Integration and Characterization

To determine the influence of the paper sensor on the mechanical properties of the fiber-reinforced lightweight structure, a blank paper was integrated into the samples. Tensile test was carried out using a universal testing machine (Zwick Roell, Z020, Ulm, Germany) according to EN ISO 527-5:1997 standard [29]. The specimens were prepared in the form of rectangular strips of size 250 mm × 25 mm × 2 mm (L × b × h). At least five specimens were tested for each configuration.

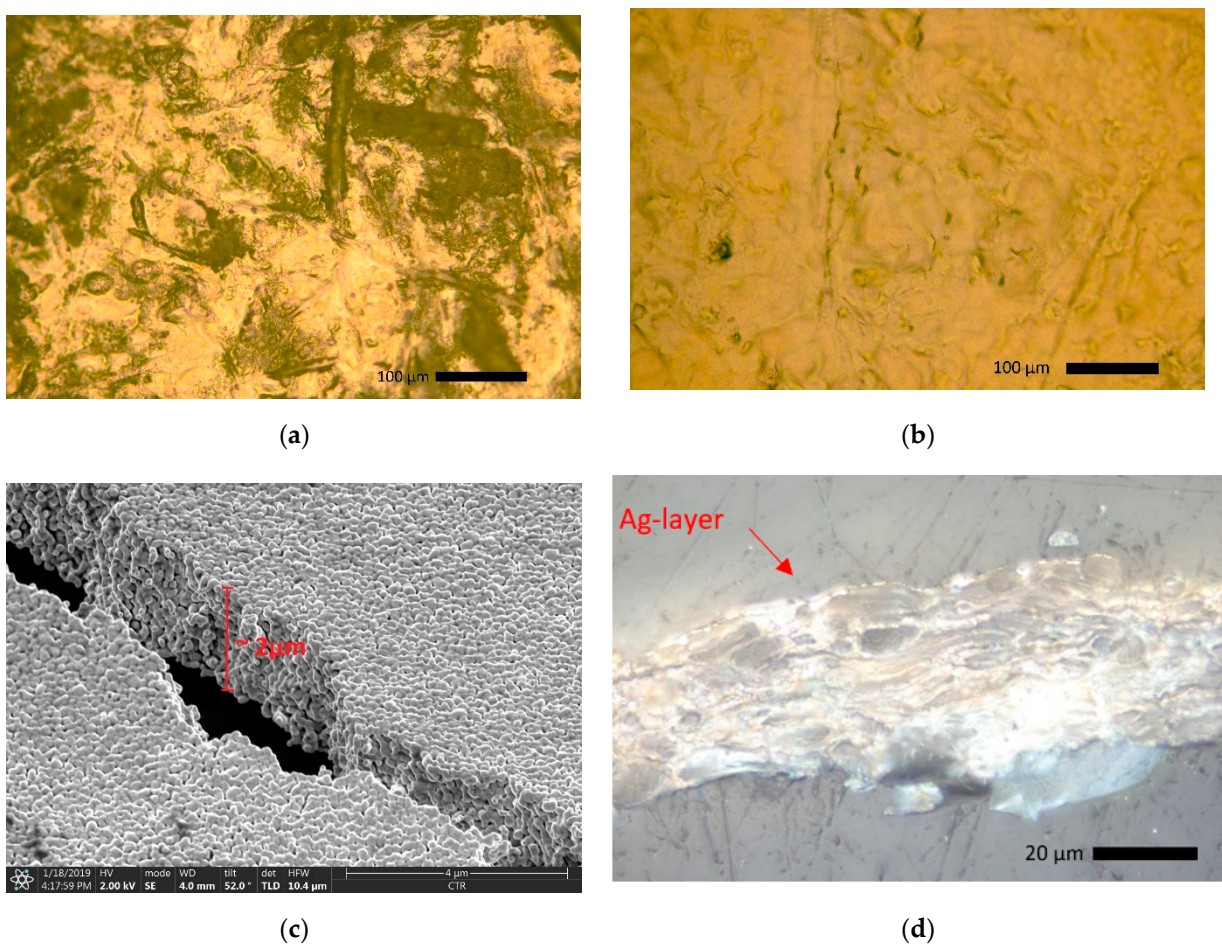

**Figure 2.** Optical microscopy image (20×) of (**a**) one and (**b**) two sintered inkjet-printed Ag-layers on the paper substrate; (**c**) SEM image of a crack in the printed layer, which reveals a layer thickness of approximately 2 μm; (**d**) optical microscopy image: cross-section of printed layer on paper substrate.

For the sensor integration, silver wires were attached to the sensors using fast set epoxy adhesive (Araldite, Huntsman International LLC, Houston, Texas, USA). To ensure a good electrical connection between the sensor and wire, silver paste (Oegussa, Vienna, Austria) was applied prior to the application of the adhesive.

Figure 3a,b. Therefore, a composite laminate was reinforced with six layers of unidirectional natural fibers, i.e., flax fibers (Eco-technilin, Valliquerville, France) of 180 g/cm$^2$ each. Bio-based epoxy (Sicomin, SR info green 810, Châteauneuf-les-Martigues, France) was used along with the hardening agent in the ratio of 100:33 (based on wt% of resin). In brief, all the materials, including natural fibers and the sensor, were arranged into the mold. Subsequently, the mold was covered with vacuum foil. Once the vacuum was created, the resin was introduced into the mold through inlets using tubing. As soon as sufficient resin was sucked into the mold, the inlets were closed, and the resin was allowed to cure at room temperature, followed by post-curing it for 16 h at 80 °C in an oven. The electrical resistance of the sensor (before and after the infusion) was monitored to study the effect of resin on the electrical properties of the sensor. Figure 4a shows a bare paper sensor before the electrical connection and the integration, while an integrated sensor sample as used for the tests is illustrated in Figure 4b.

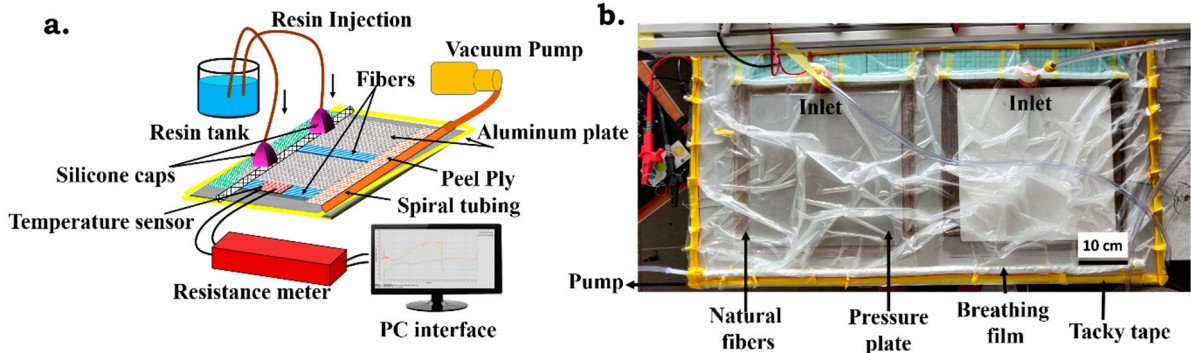

**Figure 3.** Vacuum infusion process: (**a**) schematic representation; (**b**) digital photograph.

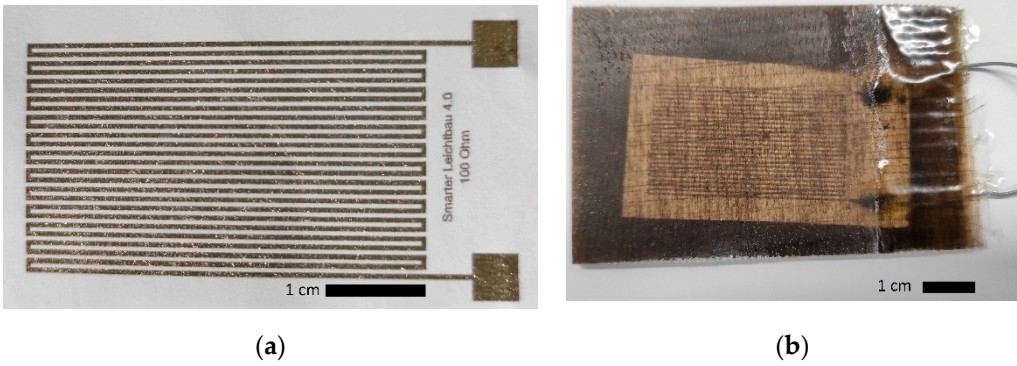

(**a**)                      (**b**)

**Figure 4.** (**a**) Printed temperature sensor on the paper substrate; (**b**) bottom view of integrated sensor sample.

The sensors were then simultaneously exposed to varying environmental conditions inside of a climatic chamber while recording the associated resistances. The sensor performances were studied in the temperature range of interest (−20 to 60 °C), which covers the range that the lightweight composites are expected to be exposed to during their useful life in the fields when being integrated into, e.g., rotor blades for small wind turbines. Due to the integration into the lightweight material, the sensor response time is expected to be significantly larger than for the bare sensor. Therefore, in the first step, the response times of the individual embedded sensors were determined by employing step response functions with an increasing step size of 20 °C in the range between −20 °C and 60 °C and vice versa.

In a subsequent test, the samples were exposed to ambient humidity levels between 20%rH and 90%rH at temperatures ranging from −20 to 60 °C in steps of 20 °C and the performance compared to a non-encapsulated paper temperature sensor. The sensors' resistances were measured every 30 s using a Keithley Digital multimeter with a measurement current of 1 mA at a pulse length of 0.002 s.

## 3. Results

### 3.1. Sensor Integration

Figure 5a shows the optical microscopy (VHX-7000 series, Keyence International, Mechelen, Belgium) images illustrating the cross-section epoxy/flax fiber composite impregnated with paper. Prior to the investigation, the surface of the composite was polished using an automated polishing machine. The thickness of the laminate was ~2 mm, and the paper thickness was 100 μm (Figure 5b). The images also indicate that the fibers and the impregnated paper fused well with the epoxy matrix. The stress-strain curves of epoxy-natural fiber composites with and without the integrated paper sheet are illustrated in Figure 6a. The ultimate tensile strength of the composite without the integrated paper was 172 MPa, while it increased to 240 MPa upon the integration of paper. The increase in

the tensile properties indicates that the impregnation of paper contributed to the reduction of fiber and surface defects, which might be due to the resin filling rich zones and by captivating voids. In addition, the interfacial interactions between the matrix and paper might have resulted in better stress transfer, consequently contributing to the augmentation of the tensile strength. This indicates that the impregnation of paper does not reduce the integrity and mechanical stability of the lightweight structure but that it even strengthens it [30,31].

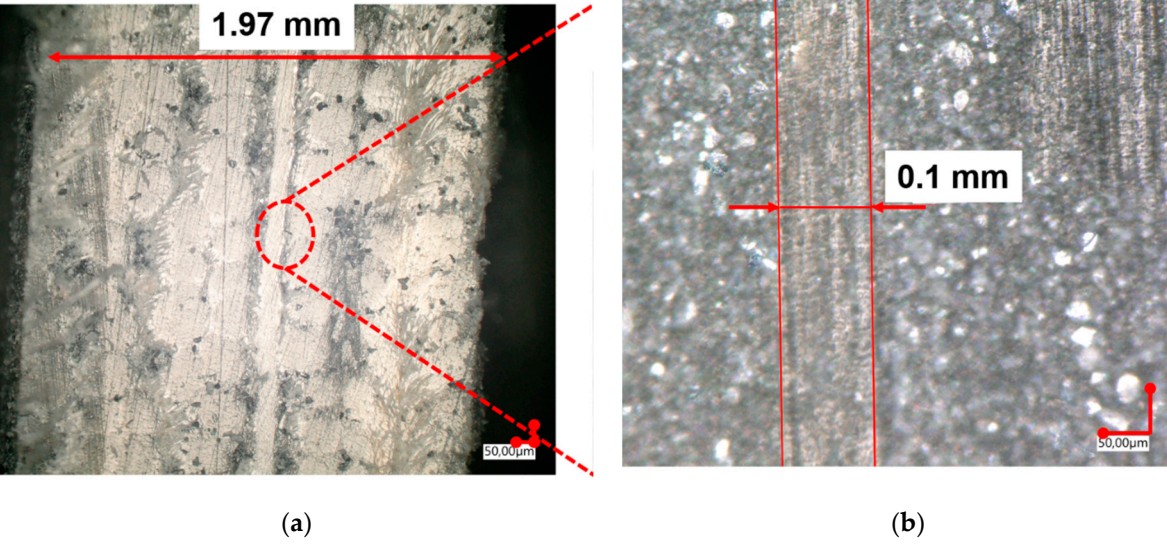

(a) (b)

**Figure 5.** (**a**) Optical microscopy image showing the cross-section of the epoxy/natural fiber composite laminate with paper; (**b**) enlarged view of impregnated paper in the composite laminate.

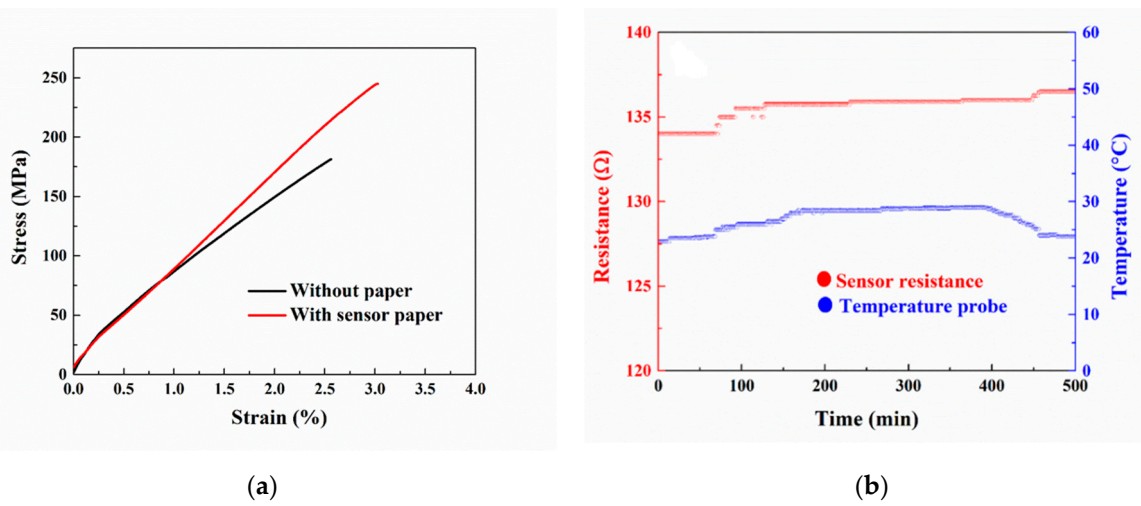

(a) (b)

**Figure 6.** (**a**) Tensile stress-strain curves of epoxy-natural fiber composite integrated with and without paper; (**b**) response of two sensors during the vacuum infusion process of epoxy-natural fiber composite.

Figure 6b shows the resistance change of one exemplary sensor during the fabrication of the composite. To affirm, a J-type thermocouple was placed next to the sensor to probe the temperature change during the process. When the uncured resin flowed over the sensors, the electrical resistances of the sensors remained unchanged. After 60 min, the sensors' resistances increased slightly, which might be attributed to the increase in the temperature of the resin because of the initiation of the exothermic curing reaction [32]. Subsequently, the sensors' resistances increased further; even after completion of the curing process, the nominal resistance values irreversibly changed, which could be attributed to

fiber swelling due to absorption as well as thermal expansion and chemical shrinkage of the epoxy during curing, leading to cracks in the printed structure.

When measuring the temperature-dependent resistance, the obtained value can be significantly affected by a local rise in temperature inside of the conductive traces due to the current flow, commonly referred to as the Joule heating effect. For the given measurement setup, the influence of Joule heating was empirically studied on the integrated sensor sample. Figure 7 illustrates this effect at a constant ambient temperature of 25 °C, employing a measurement current of 1 mA and a signal length of 2 ms at a measurement frequency of 4 Hz (Test 1); the resulting rise in resistance amounted to 0.3% of the nominal resistance $R_0$ at 25 °C. In contrast to that, Joule heating did not affect the measurement signal when the frequency was kept low (1 measurement every 30 s); therefore, in the following tests, the resistance was measured at a frequency of 0.03 Hz.

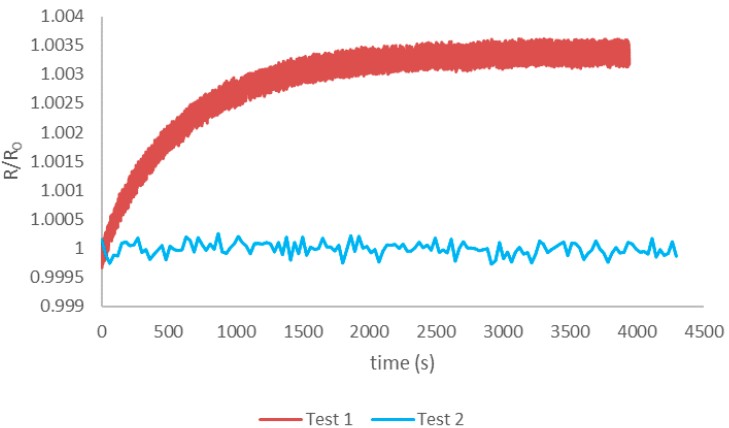

**Figure 7.** Resistance measurement over time at constant ambient temperature (25 °C) with a measurement current of 1 mA and a signal length of 2 ms at a measurement frequency of 4 Hz (Test 1) and 0.03 Hz (Test 2).

### 3.2. Thermal Sensor Characteristics

Two sensors were exposed to a temperature profile following a step response function with step sizes of 20 °C in the application-relevant range between −20 and 60 °C, as illustrated in Figure 8c. All temperature responses followed the same exponential function, regardless of the absolute temperatures and whether they were rising or falling, as exemplarily illustrated for sensor 1 in Figure 8a,b. Consequently, the time-dependent temperature T(t) can be modeled as:

$$T(t) = T_0 \times \left(1 - e^{-\frac{t}{\tau}}\right) \tag{2}$$

And:

$$T(t) = T_0 \times \left(e^{-\frac{t}{\tau}}\right) \tag{3}$$

For increasing and decreasing temperature, respectively. The response time $\tau$ is a time constant that defines the time that a sensor needs to reach 63.2% ($\frac{T(t)}{T_0} = 0.632$) of a sudden temperature change under specified conditions. The sensors reached the actual temperature (in the following, referred to as steady-state condition) within five times this time constant. Both sensors showed a similar response time around $\tau = 11$ min for a temperature change of 20 °C, which means that the steady-state was reached after $5\tau = 55$ min.

Figure 8d shows the temperature-dependent changes in resistances of the three sensors after having reached a steady state. From the graphic representation, a linear behavior was observable without any hysteresis. From this data, the TCR could be calculated according to Equation (1), as presented in Table 1.

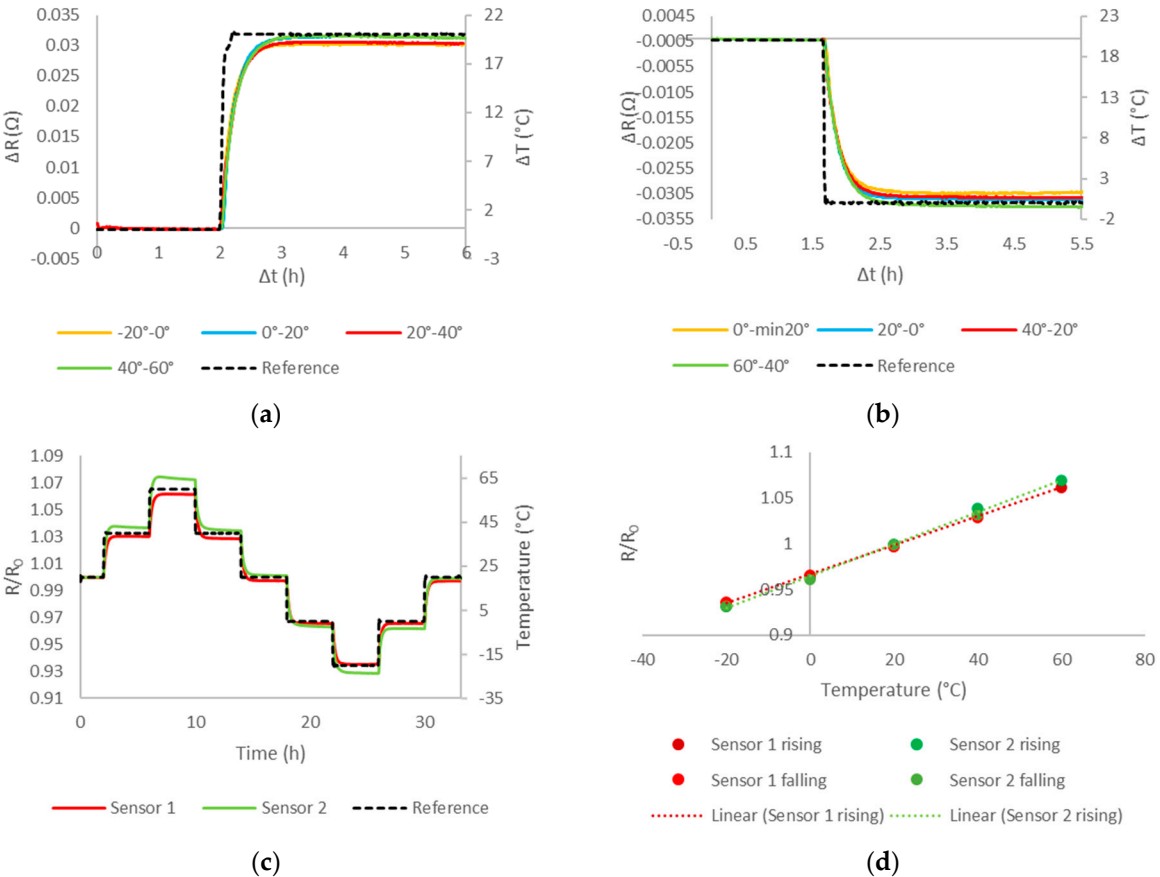

**Figure 8.** Response of sensor 1 to the step function: (**a**) rising temperature; (**b**) falling temperature; (**c**) sensor responses to temperature step response function ranging from −20 to 60 °C with a step-size of 20 °C; (**d**) temperature response at steady state (5τ).

**Table 1.** Calculated temperature coefficients of resistance (TCR) of the sensors under test and the nominal resistances $R_0$ at a temperature of 20 °C.

| Sensor | TCR in $K^{-1}$ | $R_0$ (20 °C) in $\Omega$ |
|---|---|---|
| Sensor 1 | $1.576 \times 10^{-3}$ | 200.0 |
| Sensor 2 | $1.713 \times 10^{-3}$ | 140.6 |

### 3.3. Sensor Characteristics in Humid Environments (20%rH to 90%rH)

The sensor samples were then exposed to different humidity levels. First, the temperature was kept constant at 20 °C, while the ambient humidity level was increased from 40%rH to 90%rH (40%rH–60%rH–80%rH–90%rH) and vice versa. In the following step, the temperature was increased from 20 to 60 °C in steps of 20 °C and subsequently decreased down to 0 °C. At each temperature level, the relative humidity was increased from 20%rH to 90%rH and then back to 20%rH, as illustrated in Figure 9a. As a comparison, one bare paper temperature sensor was exposed to the same cycle (Figure 9b). At 20 °C, the laminated samples did not show any remarkable response to the changes in ambient humidity, as illustrated in Figure 9c. For sensor 1, the change in average resistance due to an increase in ambient humidity from 20% to 90% at a constant temperature of 20 °C amounted to $\Delta(R/R_0) = 0.001$ (1‰), which equaled to a calculated temperature measurement error of $T_{err} = 0.69$ °C (Equation (1). At a temperature of 60 °C and when increasing the humidity level from 20%rH to 90%rH, the mean resistance change of the same sensor became larger. It equaled to $\Delta(R/R_0) = 0.0026$ (2.6‰), which accordingly resulted in a temperature measurement error of $T_{err} = 1.64$ °C. However, this effect was

reversible; after the humidity level was lowered again, the sensors recovered. To put those results into relation, the standard deviation of the sensor data at different temperature levels and humidity levels were calculated (Table 2 for sensor 1). This calculation revealed that even at 20 °C and 20%rH, the standard deviation equaled to 0.42‰ ($T_{err}$ = 0.27 °C).

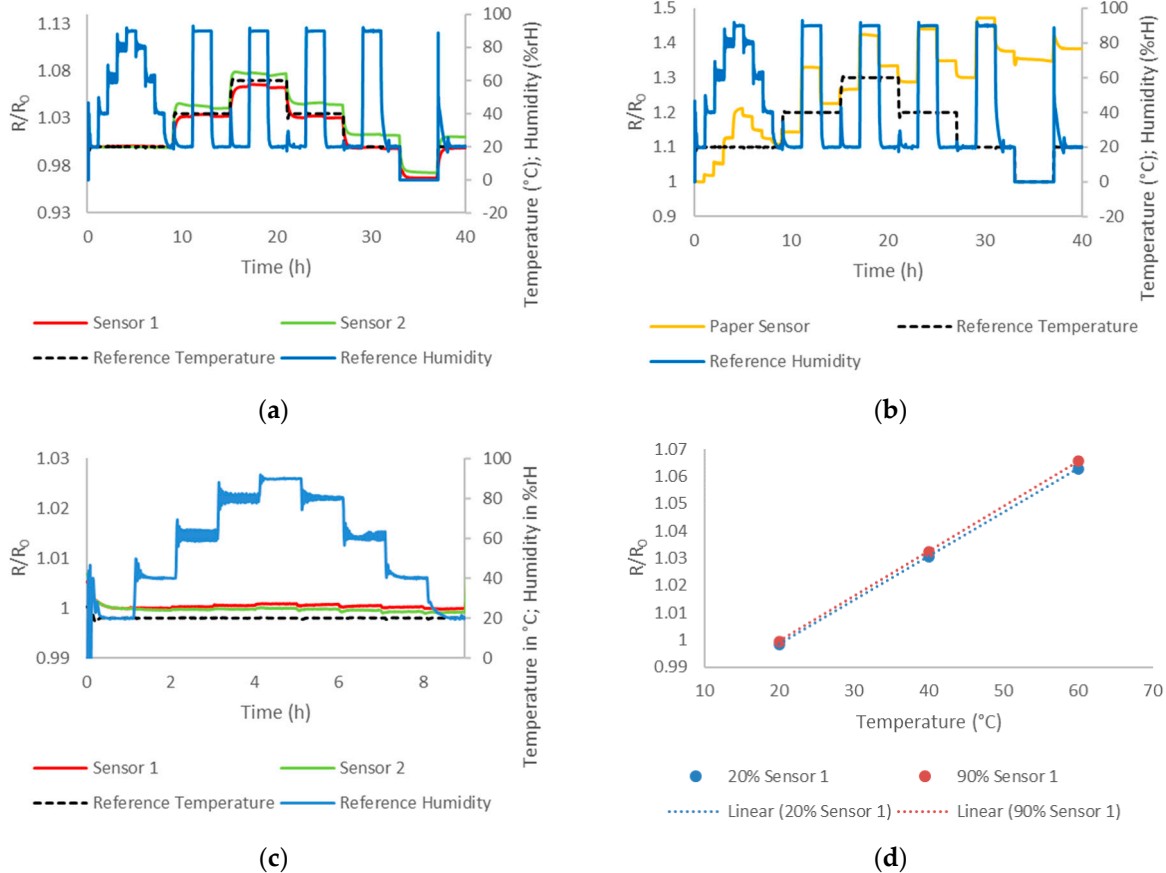

**Figure 9.** (**a**) Sensor responses of encapsulated sensors to different humidity levels at 0 °C, 20 °C, 40 °C, and 60 °C (overall test record); (**b**) sensor response of the non-encapsulated paper sensor to different humidity levels at 0 °C, 20 °C, 40 °C, and 60 °C (overall test record); (**c**) sensor responses of encapsulated sensors at humidity levels changing from 20%rH to 90%rH and vice versa at a constant ambient temperature of 20 °C; (**d**) temperature-dependent change in resistance of sensor 1 at 20%rH and 90%rH.

**Table 2.** Temperature-dependent change in resistance ΔR at different temperature and humidity levels and calculated standard deviation s of sensor 1.

| Temperature in °C | ΔR at 20%rH | s(ΔR) at 20%rH | ΔR at 90%rH | s(ΔR) at 90%rH |
|---|---|---|---|---|
| 60 | 1.0634 | 0.00045 | 1.0660 | 0.00010 |
| 40 | 1.0312 | 0.00037 | 1.0330 | 0.00021 |
| 20 | 0.9990 | 0.00042 | 1.0001 | 0.00008 |

In addition, when comparing the sensor data after temperature equilibrium was reached (5τ = 55 min) at 20%rH and 90%rH, both sensors showed a linear temperature dependence, as exemplarily illustrated for sensor 1 in Figure 9d. Compared to that, the non-laminated paper reference temperature sensor heavily responded to ambient humidity levels, as observed in previous works [15]. After increasing the relative humidity from 20%rH to 90%rH and decreasing it back to 20%rH at a constant temperature of 20 °C, the nominal resistance irreversibly increased by 10%, as illustrated in Figure 9b. The sensor

degraded even more during the course of the test, resulting in a change in resistance of around 40% by the end of the cycle.

To further study the dependence of the sensor signal on the ambient humidity, strain gauges were applied to the sample surface to monitor potential warping and deformation at a constant ambient temperature of 60 °C and humidity levels jumping from 20%rH to 90%rH. As illustrated in Figure 10, the signal of the strain gauge revealed that the sample experienced deformation due to changes in the ambient humidity level, which equally influenced the signal of the embedded temperature sensor due to geometrical deformation of the silver sensor structure (piezoresistivity).

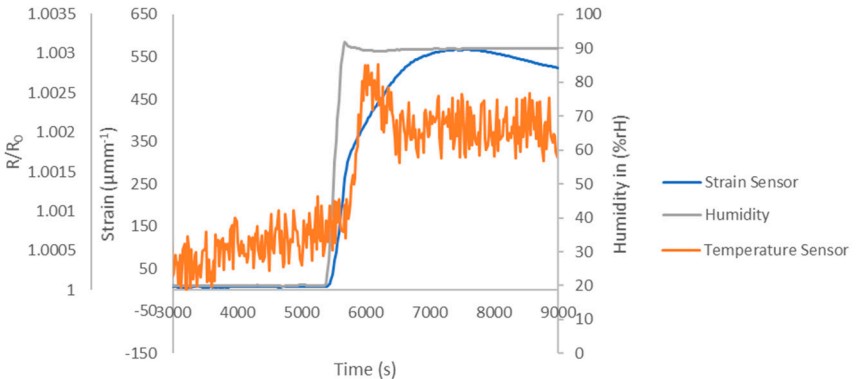

**Figure 10.** Strain measurement on the sample surface; the sample experiences deformation due to the ambient humidity, which also influences the signal of the embedded temperature sensor.

## 4. Discussion

The ultimate tensile strength of the composite increases noticeably upon the integration of paper, as illustrated in Figure 6a. This could be explained by a strong interfacial bonding between the cellulose paper and the epoxy groups. Further, the resin might have held tightly between the intervenes of the fiber, which could have provided better stress transfer and interfacial bonding. This result suggests that the integrated sensor not only monitors the condition of the composite but also improves the mechanical properties [33–35].

However, the resistance of the sensor irreversibly increases during the integration into the lightweight parts (see Figure 6b), which could be due to the resin trapped between intervenes of the paper. During the vacuum infusion process, the liquid binder is first absorbed by the paper substrate, which leads to fiber expansion and, consequently, might damage the printed structure. In addition, a piezoresistive change in resistance can be induced due to the deformation of the sample. When the curing is induced, the epoxy expands due to an exothermic reaction. During the transformation of the resin from liquid to solid, it experiences chemical shrinkage again, which is a direct consequence of crosslinking of epoxy, which leads to residual stresses and warpage. As a result, the printed sensor structure might have deteriorated to some extent, resulting in an irreversible increase of resistance [36,37].

As illustrated in Figure 7, the resistance measurement value can be significantly affected by a local rise in temperature inside of the conductive traces due to the measurement current flow, commonly referred to as the Joule heating effect. At a current of 1 mA with 2 ms pulses and a frequency of 4 Hz, the resulting rise in resistance amounts to 0.3% of the nominal resistance $R_0$ at 25 °C, which would potentially lead to a measurement error of about 2 °C. This effect can be neglected at the low measurement frequency used in the present work (1 measurement every 30 s). Therefore, depending on the application requirements, the Joule heating effect has to be taken into account, and, if necessary, different measurement parameters (current, pulse duration, or frequency) might need to be employed. Compared to the results from [15], the samples show a large response time of $\tau = 11$ min for a temperature change of $\Delta T = 20$ °C, as illustrated in Figure 8a,b, which

can be attributed to the thermally insulating properties of the fiber-reinforced lightweight embedding material. This offers novel opportunities for employing the printed sensors as a tool to determine the thermal properties of the embedding material as part of future material optimizations. In addition, the response time of the sensor can neither be considered as a limitation for the practical application task of structural health monitoring, as proposed in the present work, as ultimately the temperature inside of the material is supposed to be monitored. This can be exploited for conducting highly targeted measurements of the actual material temperature at defined device positions and material depths.

The calculated TCR of the sensors under test ranges from $1.576 \times 10^{-3}$ K$^{-1}$ to $1.713 \times 10^{-3}$ K$^{-1}$ for sensor 1 and sensor 2, respectively. This is due to the differences in the nominal resistances (140 $\Omega$ to 200 $\Omega$), which can be explained by manufacturing-related variations. Although both sensors are manufactured using the same process, they are not fabricated as one single batch under the exact same conditions. Changes in the ink composition over time, such as nanoparticle agglomeration and evaporation of solvents, can have an impact on the printing results. Furthermore, the printhead is prone to degradation over time, especially when not being operated continuously, leading to partial clogging of nozzles, resulting in different drop sizes and, consequently, affecting the amount of conductive ink that is deposited. Last but not least, the high porosity and fibrousness of the used paper substrate lead to poor reproducibility, as discussed for the used paper type in [17]. Already in previous works, the bare temperature sensors on the same substrate show large variations of the nominal resistances of around 15% [15]. Furthermore, the sensors experience increases in resistance due to the absorption and thermal deformation of the epoxy during curing. This implies that each sensor would have to be calibrated individually to gain absolute measurement results for structural health monitoring in the fields. While this would not always be needed, the printing processes used for prototyping in research cannot be compared to well-controlled industrial manufacturing conditions that are achievable in high throughput production lines. Furthermore, industrial vacuum infusion processes for the fabrication of lightweight parts also provide a higher level of reproducibility regarding the amount of binder used and the processing temperature. To put the results of this work into perspective, previous publications reported TCR values for printed silver lines ranging from $6.52 \times 10^{-4}$ K$^{-1}$ [23] to $2.19 \times 10^{-3}$ K$^{-1}$ [20], which indicates that $\alpha$ is rather dependent on the individual processing conditions than on the metallic sensing material used. However, the results from the present work are well aligned with the TCRs obtained in [15], which lie between $1.630 \times 10^{-3}$ K$^{-1}$ and $1.705 \times 10^{-3}$ K$^{-1}$.

The results from the tests in a humid environment indicate that the paper-based sensors are almost insensitive to changes in ambient humidity due to the embedding into fiber-reinforced lightweight materials, as illustrated in Figure 9a. The tests are conducted at temperatures above freezing point, as only very little water is dissolved in the air below 0 °C, for which an impact on the sensor performance is not expected. Furthermore, this would be outside the operating range of the used climatic chamber. The results for sensor 1 reveal that a change in the ambient humidity level from 20%rH to 90%rH at 20 °C and 60 °C leads to temperature errors of $T_{err}$ = 0.69 °C and $T_{err}$ = 1.64 °C, respectively. Since this effect is reversible, it might be explained by piezoresistivity. Strain measurements using commercial strain gauges on the sample surface indeed indicate that the sample mechanically deforms in humid environments (Figure 10). However, the standard deviation at steady-state and a constant humid environment (20%rH) and temperature (20 °C) results as well in a temperature measurement error of $T_{err}$ = 0.27 °C. To put those results into relation, a commercial Pt100 temperature sensing element of class C might have a temperature tolerance of around 1.2 °C at 60 °C, as specified by DIN EN 60751:2009-05 [38]. It can be assumed that those limitations in measurement accuracy are negligible for the proposed application of structural health monitoring of natural fiber-reinforced lightweight materials during their useful life.

The bare paper-based temperature sensor is shown to be extremely sensitive towards humidity, as already reported in [15] and illustrated in Figure 9b. Since this effect is irre-

versible, it might be explained by fiber expansion due to the absorption of liquids, leading to cracks in the printed layer. Hence, the results of this work indicate that embedding the sensors can be a highly effective method to avoid these degradation mechanisms, paving the way towards extremely low-cost robust paper-based sensors for humid environments.

## 5. Conclusions

In this work, low-cost inkjet-printed temperature sensors on the paper substrate were fully integrated into natural fiber-reinforced lightweight components. Subsequently, the samples were exposed to varying relevant temperature ($-20$ to $60\,^{\circ}$C) and humidity (20%rH to 90%rH) conditions inside of a climate chamber.

The integration of the sensors had the purpose of providing a sustainable solution for structural health monitoring. In addition, it improved the mechanical integrity and stability of the lightweight part, as indicated from the stress-strain curves of epoxy-natural fiber composite integrated with and without paper. The results also showed that all sensors under test remained functional after the vacuum infusion process; however, the nominal resistance increased irreversibly, which might be attributed to fiber swelling due to absorption, as well as thermal expansion and chemical shrinkage of the epoxy, which caused deterioration of the printed structures.

When being exposed to varying temperature and humidity conditions inside of a climate chamber, both sensors showed a linear temperature dependence and no hysteresis in the temperature range of interest ($-20$ to $60\,^{\circ}$C) with a TCR ranging from $1.576 \times 10^{-3}$ K$^{-1}$ to $1.705 \times 10^{-3}$ K$^{-1}$. The results from the tests in a humid environment indicated that the paper-based sensors had become almost insensitive to changes in ambient humidity after embedding them into fiber-reinforced lightweight composites.

The paper-based sensors were shown to be suitable for the integration into natural fiber-reinforced biopolymer-based lightweight composites, creating a potential platform for sustainable structural health monitoring. The complete integration of the devices, making them an inherent part of the composite material, to monitor could be considered as a highly innovative approach. Nonetheless, the monitoring of temperature alone did not provide sufficient information on the structural health status of the components; still, invaluable sensing data for a deeper understanding of degradation modes due to environmental influences could be obtained. In addition, the use of wired external devices for the readout of sensor data could not be considered as convenient for the proposed application. This drawback becomes particularly relevant when the monitoring of mobile components, such as rotor blades for small wind turbines, etc., is desired. Therefore, as part of future works, wireless readout options will be studied, as well as the applicability of different additively manufactured sensors; for e.g., humidity and strain sensing to obtain further valuable performance parameters of smart, sustainable, and environmentally compatible lightweight composite materials for the future.

**Author Contributions:** Conceptualization, J.Z. and L.R.; methodology, J.Z., M.K., and L.R.; formal analysis, J.Z., M.K., L.R., H.L., and J.K.; data curation, J.Z. and M.K.; writing—original draft preparation, J.Z., M.K., and L.R.; writing—review and editing, J.Z., M.K., L.R., H.L., and J.K.; supervision, H.L. and J.K.; project administration, L.R. All authors have read and agreed to the published version of the manuscript.

**Funding:** This work has been conducted as part of the research project Smarter Leichtbau 4.1 funded by the European Regional Development Fund (ERDF).

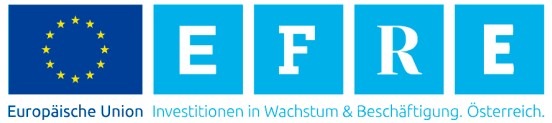

**Institutional Review Board Statement:** Not applicable.

**Informed Consent Statement:** Not applicable.

**Data Availability Statement:** Not applicable.

**Conflicts of Interest:** The authors declare no conflict of interest.

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
