# Peer review of "Low-Cost Inkjet-Printed Temperature Sensors on Paper Substrate for the Integration into Natural Fiber-Reinforced Lightweight Components"

_chemosensors, doi:10.3390/chemosensors9050095_

Round 1

Reviewer 1 Report

The authors present here the development and characterization of inkjet-printed temperature sensors on paper substrate. The work is interesting and fulfills the scope of the journal. Nevertheless, some drawbacks should be addressed before accepting the work. Please attached find my comments: 

1.    Title. The current title mentions the use of these paper sensors as a part of structural Health Monitoring. Nonetheless, the authors did not develop this application in work. Authors should reconsider a new title that better describes their work.
2.    Abstract. Could you please more highlight the uniqueness of your study here to make the abstract more attractive? 
3.    Introduction that serves here as a theoretical background of the study is poorly developed. In my opinion, the current version is not depth is rather descriptive. Please expand here. 
4.    The manuscript is not clear about the aim of the study. The only mention is a previously published paper by some of the co-authors. This information should be right in the Introduction. 
5.    Some figures (Fig 7b, Fig 8 and 9) are illegible in their actual size and resolution. Please, consider improve their content. 
6.    Results reporting the characterization are properly and deeply describe; nevertheless, they fail about the suggested application related to health monitoring.
7.    The conclusion is descriptive, which is good, but I would expect to read more about consequences, i.e., how might be proposed solution helpful. I also think in such a study, some space should be devoted to factors that affect the results. Could you please expand in the concluding part on these limitations of your study (concerning methodology, settings, materials, etc.)?
8.    As was mentioned by the authors, sensors were build following the same methodology reported in 12. Consulting this paper now presents a complete characterization using humidity and temperature variables, but the work is not associated with health monitoring. Is it possible that the authors explain this important point in the work?
9.   Finally. Please correct the reference format in the text according to the journal template. Figure numeration and captions should do not use a colon.

Author Response

Dear Reviewer,

Please see our reply in the attached file.

Sincerely,

Johanna Zikulnig

Reviewer 2 Report

From the point of view of a temperature sensor based on a printed pattern, I disagree with the practical applications. The resistance change is very small (less than 1% resistance change for 10 degree). The is not effective sensing method for temperature. Also, the temperature in the line can rise locally due to the current flow (joule heating) during resistance measurement, which is independent of the environmental temperature conditions. This joule heating effect during the resistance measurement should be fully discussed. If the very low current is used for the measurement, then the measurement gain for the sensing might be also reduced. If there is any trade off relationship, the information should be added.

         Nonetheless, printing on paper for conductive lines is worth investigating. The printed nanoparticle ink requires a sintering process to achieve adequate conductivity. The paper can be damaged or deformed during the sintering process. When I first saw the manuscript title, I expected the extensive discussion on the issues of Ag ink printing on papers. However, I cannot find detailed information about the Ag ink and the sintering process. In addition, more investigation into the thickness of the printed pattern is needed by examining the cross-sectional portion of the line on the paper. The ink can be penetrated into paper fiber and the resistivity can be affected in the case of paper. So, the resistivity of the printed line should be provided. Also, paper is flexible and the metal lines can be damaged during the bending process. These effects should be discussed. In conclusion, I do not recommend the journal publication unless more analysis (including sintering process effects, bending effects, resistivity, microscopic analysis of cross-section of printed lines etc..) is added to the revised manuscript. 

Author Response

(The authors gave the same response as above.)

Reviewer 3 Report

Dear Authors,

The performed study is very well presented, formally the manuscript covers all the usual chapters, the methods as well as the results are clearly described as continuation of your previous paper. Having in mind there are no changes to be done regarding your manuscript, let me point to several minor suggestions:

When mentioning the temperature frames, in lines 21, 62, 64 and 321, I'd suggest you use the expression "to" instead of "-" just as you did in line 130, since you start with negative temperature value, using "-" might cause confusion. The same may apply for rH, line 64.

In lines 113 and 211 I'd omit the word "see" in bracelets, Figure 4 ....  and Table 2. ... would do.

In line 136 when you mention ... different temperatures ... you might wish to define which temperatures you had in mind, if it is the same range from -20°C to 60°C, it would not harm to repeat it.

In line 293 please adjust the inscription to either letters or symbols, i.e zero degree Celsius or 0°C ...

The above mentioned are just my suggestions.

Author Response

Dear Reviewer,

Thank you for taking the time to review our manuscript and the positive feedback which is highly encouraging also for our future scientific work.

We’ve adapted the manuscript according to your suggestions.

Kind regards,

Johanna Zikulnig

Round 2

Reviewer 1 Report

Thank you for your updates. The manuscript became to be more strong and more self-contained

Reviewer 2 Report

Acceptable for publication.